

# Construction and Application of a Pollen Emissions Model based on Phenology and Random Forests

Jiangtao Li[a,b], Xingqin An[a,*], Zhaobin Sun[a], Caihua Ye[c], Qing Hou[a], Yuxin Zhao[a], Zhe Liu[a]

[a] State Key Laboratory of Severe Weather, Chinese Academy of Meteorological Sciences, Beijing, 100081, China

[b] Department of Atmospheric and Oceanic Sciences & Institute of Atmospheric Sciences, Fudan University, Shanghai, 200438, China

[c] Meteorological Service Center of Beijing Meteorological Bureau, Beijing, 100089, China

[*] Corresponding author: Xingqin An

Email address: anxq@cma.gov.cn

**Abstract:** In recent years, the intensification of global climate change and environmental pollution has led to a marked increase in pollen-induced allergic diseases. This study leverages 16 years of continuous pollen monitoring data, alongside meteorological factors and plant functional type data, to construct a pollen emissions model using phenology and random forests (RF). This model is then employed to simulate the emission characteristics of three primary types of autumn pollen (Artemisia, Chenopod, and total pollen concentration), elucidating the emission patterns throughout the seasonal cycle in Beijing. Phenology and RF precisely simulate the start and end day of year of pollen, as well as the annual pollen production. There are significant spatiotemporal differences among the three types of pollen. On average, pollen dispersal begins around August 10, peaks around August 30, and concludes by September 25, with a dispersal period lasting approximately 45 days. Furthermore, the relationship between pollen emissions and meteorological factors is investigated, revealing that temperature, relative humidity (RH), and sunshine hours (SSH) significantly influence annual pollen emissions. Specifically, temperature and RH exhibit a strong positive correlation with annual pollen emissions,





while SSH shows a negative correlation. Different pollen types display varied
responses to meteorological factors. Finally, the constructed pollen emissions model is
integrated into RegCM and validated using pollen observation data, confirming its
reliability in predicting pollen concentrations. This study not only enhances the
understanding of pollen release mechanisms but also provides scientific evidence for
the selection and planting of urban greening plants.
**Keywords**: Pollen Emissions Model, Phenology, Random Forests, RegCM

## 36   1. Introduction

Pollen are microscopic particles, typically ranging from 5 to 100 micrometers in
diameter, released by plants to transfer male genetic material for reproduction. These
particles, significant allergens, disperse into the atmosphere via wind, contributing to
atmospheric particulate matter, interacting with clouds and radiation, and playing a
pivotal role in plant fertilization and gene dissemination (Damialis et al., 2011; Lei et
al., 2023). Additionally, pollen is linked to allergic diseases such as allergic rhinitis
and asthma and may even elevate the risk of gastrointestinal and neurological
disorders (Guzman et al., 2007; Krishna et al., 2020; Chen et al., 2021; Stas et al.,
2021). In China, the incidence of pollen allergies has surged from 5 % to 17.8 % and
continues to rise rapidly (Lou et al., 2017). Pollen-induced respiratory allergic
symptoms, such as allergic rhinitis (AR), affect up to 30 % of the global population,
particularly children under 18 (Mir et al., 2012; Wang et al., 2016; Zhang and Steiner,
2022; Zhao et al., 2023). It is generally believed that these respiratory allergic
diseases are more prevalent in developed countries (Emanuel, 1988; Ibrahim et al.,
2021). However, the International Study of Asthma and Allergies in Childhood
(ISAAC) global reports indicate that these diseases are equally or even more prevalent
in some developing countries compared to developed ones (Asher et al., 2006; Mallol
et al., 2013). Children, as a vulnerable population, are particularly susceptible to AR
and its complications (Cingi et al., 2017). Without effective early intervention, allergic
symptoms in children can persist throughout their lives, imposing a substantial
economic burden on families and healthcare systems (Ahmed et al., 2018) and





potentially posing a life-threatening risk (Schmidt, 2016). In China, a densely
populated developing country, the proportion of pediatric allergic diseases within the
spectrum of childhood illnesses is increasing annually, leading to significant
economic and health losses due to medical expenses, impacts on human life, and
premature death. Furthermore, since pollen release is closely linked to environmental
factors, climate change may influence pollen release, thereby affecting the incidence
of allergic diseases (Wang et al., 2018; Bishan et al., 2020). In recent decades, the
pollen season has exhibited a trend of becoming longer and more intense, which may
exacerbate the conditions of allergic rhinitis and asthma (D'Amato et al., 2016; Lake
et al., 2017a; Aerts et al., 2020; Kurganskiy et al., 2021)..
With the improvement in living standards and heightened health awareness,
airborne pollen diseases, such as hay fever, have garnered widespread attention. As a
typical seasonal epidemic (Yin et al., 2005; Lei et al., 2023), hay fever significantly
impacts global health. Existing studies have demonstrated that the incidence of
airborne pollen diseases is closely associated with the concentration of airborne
allergenic pollen, particularly during peak pollen seasons (Frei and Gassner, 2008;
Bastl et al., 2018; Kurganskiy et al., 2021). Due to the regional nature of airborne
pollen, the types and concentrations of pollen vary geographically. Although the
annual variation trend of total pollen amount generally exhibits a similar bimodal
pattern, increasing annual climatic variability amidst global warming has led to
significant changes in the pollen seasons of various plants, with discrepancies of more
than 20 days in some years. This variability poses practical challenges for conducting
pollen monitoring research and providing public meteorological services (He et al.,
2001; Gu and Liao, 2003; Bai et al., 2009; Lei et al., 2023). Therefore, studying
pollen concentration and distribution is crucial for understanding the pathogenesis of
airborne pollen diseases, conducting effective pollen monitoring research, and
delivering accurate public meteorological services.
However, compared to regions such as Europe and the United States, China faces
significant challenges in pollen monitoring due to fewer monitoring stations, shorter
monitoring histories, and a lower prevalence of automated facilities. These limitations



have resulted in China's pollen simulation research remaining primarily at the level of
simple statistical methods, focusing only on basic statistical studies of the impact of
meteorological conditions on pollen concentration. In contrast, numerical models are
rarely employed for regional simulation of pollen concentration. This situation
reflects the relative lag in China's pollen monitoring and research system, hindering a
deeper understanding of pollen dispersion patterns and the scientific study of related
health issues (Wu et al., 2011; Meng et al., 2016; Guan et al., 2021; Gao et al., 2022).
Although numerical models play a crucial role in simulating pollen concentration,
they require a clear understanding of pollen emissions. Pollen emissions are
influenced not only by meteorological factors but also by vegetation types, land use
changes, and human activities (Sofiev et al., 2006; Wozniak and Steiner, 2017; Zhang
and Steiner, 2022; He et al., 2023; Lei et al., 2023). Particularly in the context of
accelerated urbanization, the selection and layout of urban greening plants have a
significant impact on pollen emissions. The complex interactions of these factors pose
significant challenges to accurately simulating pollen emissions.
Therefore, this study constructs a pollen emissions model for the Beijing area,
leveraging pollen concentration and meteorological monitoring data, combined with
pollen phenology and the RF algorithm. It conducts a simulation study on the
emission phenology of three types of pollen in Beijing (Artemisia, Chenopod, and
total pollen concentrations) to calculate the pollen emissions potential. The study also
investigates the seasonal and spatiotemporal distribution characteristics of pollen in
Beijing and its potential correlations with meteorological factors and climatic
conditions. Additionally, the constructed pollen emissions parameterization method is
applied to the RegCM and evaluated for accuracy using 15 years of pollen
observation data. This comprehensive study will enhance the understanding of pollen
sources, provide innovative guidance for the selection and planting of greening plants,
and promote sustainable development in ecological protection and urban planning.
**2. Methodology**
*2.1 Model description*





2.1.1 Parameterization method for pollen emissions
This study's pollen emissions potential integrates geographical parameters,
vegetation types, and meteorological data, and incorporates autumn pollen phenology
and RF to enhance the simulation of pollen phenology. This approach is used to
predict pollen concentration and distribution within the seasonal cycle. The specific
calculation formula is as follows:
$$E_i(t) = f_i \bullet p_{annual,i} \bullet e^{-\frac{(t-\mu)^2}{2\delta^2}} \tag{1}$$

In the formula, $E_i(t)$ represents the pollen emissions potential for pollen type $i$ on
day $t$ of the year (DOY), $t$ represents a specific day of the year, and $i$ represents the
$i$-th type of pollen. $f_i$ represents the vegetation land cover fraction, which is the
percentage of different vegetation types within a unit area, measured in %. $P_{annual,i}$
represents the production factor of the $i$-th vegetation type, which is the number of
pollen grains released during the pollen season, measured in *Grain m$^{-2}$ year$^{-1}$*. In this
study, $P_{annual,i}$ is calculated based on the RF algorithm (Sect. 2.1.3). $e^{-\frac{(t-\mu)^2}{2\delta^2}}$
represents the phenological evolution of pollen emissions, controlling the pollen
release process. The formula indicates that pollen emissions during the pollen season
follows a Gaussian distribution, where $\mu$ and $\delta$ are the mean and standard
deviation of the Gaussian distribution. These parameters are calculated from sDOY
and eDOY of the pollen season, as follows:

$$\mu = \frac{sDOY + eDOY}{2} \tag{2}$$
$$\delta = \frac{eDOY - sDOY}{a} \tag{3}$$

In this context, sDOY and eDOY are optimized using autumn pollen phenology
(Sect. 2.1.2). The parameter $a$ represents a fitting parameter that explains the
conversion between the empirical phenological dates based on pollen count thresholds
and the equivalent width of the emission curve. In this study, the value of $a$ is set to 4.
This equation can be applied to a specific type of pollen or to the calculation of
pollen concentration over the entire pollen season, depending mainly on the land





cover type. The emission can be calculated offline using this equation or applied in
online calculations.
2.1.2 Autumn pollen phenology model
In this study, we used three different calculation methods ($Rs_1$, $Rs_2$, $Rs_{sig}$) for the
autumn phenology model to simulate sDOY and eDOY of autumn pollen (Meier &
Bigler, 2023). Each model is related to temperature and SSH. The specific calculation
formulas are as follows:

$$Rs_1 = \begin{cases} (T_{base} - T_i)^x \times (L_i / L_{base})^y, & T_i < T_{base} \wedge L_i < L_{base} \\ 0 & , T_i \geq T_{base} \vee L_i \geq L_{base} \end{cases} \tag{4}$$


$$Rs_2 = \begin{cases} (T_{base} - T_i)^x \times (1 - L_i / L_{base})^y, & T_i < T_{base} \wedge L_i < L_{base} \\ 0 & , T_i \geq T_{base} \vee L_i \geq L_{base} \end{cases} \tag{5}$$


$$Rs_{sig} = \frac{1}{1 + e^{a(T_i \times L_i - b)}} \tag{6}$$


$$\sum_{t_0}^{t_n} Rs_i \geq Y \tag{7}$$

In the above equations, $Rs_1$, $Rs_2$ and $Rs_{sig}$ represent three different autumn
phenology model categories. $T_i$ and $L_i$ represent the temperature and SSH on a given
day, respectively, while $T_{base}$ and $L_{base}$ represent the thresholds for temperature and
SSH, respectively. In the $Rs_1$ and $Rs_2$ models, when the temperature and SSH are
below the threshold or the date exceeds a fixed DOY, Rs starts accumulating. In the
$Rs_{sig}$ model, temperature and SSH accumulate inversely in an exponential form. The
day $t_n$, when the cumulative amount exceeds the threshold $Y$, represents the final
simulated pollen start/end date. $t_0$ represents the start day of accumulation, which is
the first day when $T_i < T_{base}$ and $L_i < L_{base}$. The parameters that need to be adjusted are $Y$,
$T_{base}$, $L_{base}$, $x$, $y$ and *start_day*. In this study, the simulated annealing algorithm is used
for parameter adjustment. The principle of the simulated annealing (SA) algorithm is
to simulate the random optimization process of the annealing process in solid-state
physics, which can accept non-optimal solutions with a certain probability to avoid
falling into local optima and eventually achieve the global optimum.
2.1.3 Random Forests



Random Forests (RF) is an ensemble learning algorithm introduced by Breiman
(2001) for classification and regression tasks. This algorithm enhances model
prediction performance and robustness by constructing multiple decision trees and
combining their outputs. The core principle involves drawing multiple sample sets
with replacement from the original training set, training a decision tree for each
sample set, and randomly selecting a subset of features at each node split to reduce
correlation between the trees. Ultimately, RF generates the final prediction by
averaging (for regression) or voting (for classification) the outputs of these trees. The
advantages of this method include high prediction accuracy, strong resistance to
overfitting, suitability for high-dimensional data, and efficient training processes. The
RF algorithm has been widely applied across various fields (Virro et al., 2022; Li et
al., 2023; Chen et al., 2024; Valipour Shokouhi et al., 2024).
In this study, the RF algorithm is employed to simulate annual pollen production.
Each pollen dataset is divided into training and testing sets in a 4:1 ratio, with the
training set used for model training and the testing set for accuracy validation.
Additionally, a grid search with cross-validation is applied to optimize the
hyperparameters of each estimator. Key parameters for RF adjustment include
n_estimators, max_depth, min_samples_split, and min_samples_leaf. Hyperparameter
optimization is a crucial step in enhancing model performance.
*2.2  Data*
*2.2.1  Observed pollen concentrations*
The daily pollen concentration data were collected from six monitoring stations
in Beijing: Changping (CP), Chaoyang (CY), Fengtai (FT), Haidian (HD),
Shijingshan (SJS), and Shunyi (SY), as shown in Fig. 1. The monitoring period
spanned from April to October each year from 2006 to 2021, covering the main pollen
season in Beijing. The gravitational settling method (Unit: $10^3$ Grains m$^{-2}$ d$^{-1}$) was
used for monitoring. The pollen concentration data included Total Pollen
Concentration (the sum of pollen concentrations from all taxa, abbreviated as TotalPC)
and the concentrations of pollen from 10 common allergenic plants. These species
included trees such as Pine, Poplar, Birch, Cypress, Ash, and Elm, as well as weeds



like Artemisia, Chenopod, Humulus, and Amaranthus. Although autumn pollen
concentrations are lower compared to spring, autumn weed pollen has a higher
allergenic potential (Zhao et al., 2023). Therefore, this study focuses on the analysis
of autumn weed pollen. Due to significant data gaps in the pollen concentration of
specific species, we selected only the data that were more complete and of higher
allergenic potential, specifically Artemisia, Chenopod, and TotalPC. Table 1 provides
basic information, such as the number of effective sample years for these three types
of pollen across the six stations.
To prevent anomalies in the data, we excluded outliers in the pollen
concentration data for each species and any data points where the concentration
exceeded the 99th percentile. Furthermore, we applied a 5-day moving average to the
pollen monitoring data to smooth it. This approach not only eliminates noise from the
data (Li et al., 2019; Li et al., 2022) but also mitigates the influence of daily
meteorological changes and advection diffusion on daily pollen emissions (To further
analyze the impact of key factors such as meteorological factors and advection
diffusion on daily pollen emissions, we used the RegCM in Sect. 3.3. This model
accurately reflects the effects of daily meteorological factors such as temperature,
precipitation, humidity, and wind speed on pollen emissions while also describing key
physical processes such as advection diffusion, convective transport, and dry and wet
deposition, thus providing a comprehensive analysis of the behavior of pollen in the
atmosphere). This smoothing process allows us to more clearly explore the daily
variation trends of pollen.
Additionally, to better simulate the temporal and spatial distribution of pollen
during the autumn pollen period, we defined the autumn pollen period based on
observed pollen concentration data as DOY 215<DOY<280. Subsequently, we
determined the Start Day of Year (sDOY) and End Day of Year (eDOY) for the
autumn pollen period for each station and year by identifying the day of year at which
the cumulative pollen concentration reached 5 % (start) and 95 % (end) of the total for
that period (Khwarahm et al., 2017; Li et al., 2019; Li et al., 2022).





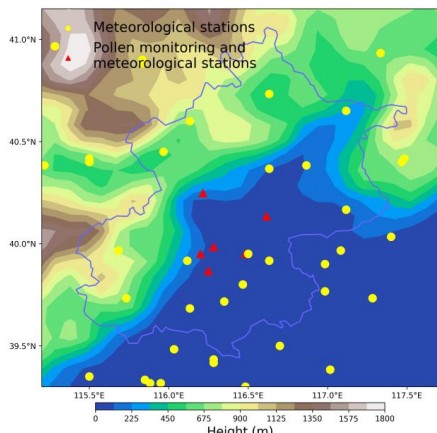


Fig. 1. Distribution map of geopotential height, pollen observation stations (triangle), and
meteorological monitoring stations (circle) in Beijing area
Table 1 Explanation of effective sample years for pollen monitoring stations in Beijing

(2006-2021)

| Station | Effective Sample Years / Year | | |
|---|---|---|---|
| | Artemisia | Chenopod | TotalPC |
| CP | 16 | 16 | 16 |
| CY | 13 | 13 | 13 |
| FT | 10 | 8 | 15 |
| HD | 0 | 0 | 8 |
| SJS | 11 | 11 | 16 |
| SY | 12 | 9 | 16 |
| Total | 62 | 57 | 84 |

To better simulate sDOY and eDOY for pollen, this study first applied the
Gaussian model to the autumn pollen data of each station and year. The Gaussian
model was chosen for its effectiveness in capturing peaks in time series data, which
are often reflected in pollen concentration data. Taking the CP station as an example,
Gaussian fitting distribution was performed on the autumn Artemisia, Chenopod, and
TotalPC for 2006-2021 (Supplementary Fig. S1-S3). The results indicated that the
autumn pollen concentration exhibited a significant Gaussian distribution, confirming
that the Gaussian model could aptly fit the time series changes of autumn pollen.



Therefore, by Gaussian fitting the pollen concentrations of each station, the autumn
pollen sDOY and eDOY under the Gaussian model simulation were further
determined. Comparing the sDOY and eDOY derived from observed pollen
concentration data with those obtained via Gaussian model simulation
(Supplementary Fig. S4), we found a high correlation coefficient (R) and a low root
mean square error (RMSE) between the two. Thus, the sDOY and eDOY obtained
from Gaussian model simulation were utilized to study the autumn pollen phenology.
*2.2.2 Meteorological observation and land cover data*

The meteorological data for this study were sourced from the China Surface

Climate Daily Dataset, encompassing observations from all benchmark and basic
meteorological stations in China. Specifically, we utilized data from 66 valid
meteorological stations in Beijing and its surrounding areas (39-41.5° N, 115-118° E)
covering the period from 2006 to 2020 (Fig. 1). This dataset includes meteorological
observations corresponding to the pollen monitoring stations (our meteorological data
extends only up to 2020). The variables incorporated in this study comprise average
temperature (TEM_Avg), maximum temperature (TEM_Max), minimum temperature
(TEM_Min), sunshine hours (SSH), station altitude (Alti), average pressure
(PRS_Avg), maximum pressure (PRS_Max), minimum pressure (PRS_Min),
maximum wind speed (WIN_S_Max), extreme wind speed (WIN_S_Inst_Max),
average 2-minute wind speed (WIN_S_2mi_Avg), ground surface temperature
(GST_Avg_Xcm, X=5, 10, 15, 20, 40, 80, 160, 320cm), average ground surface
temperature (GST_Avg), minimum ground surface temperature (GST_Min),
maximum ground surface temperature (GST_Max), average relative humidity
(RHU_Avg), minimum relative humidity (RHU_Min), average vapor pressure
(VAP_Avg), precipitation from 20:00 to 20:00 (PRE_Time_2020), and precipitation
from 08:00 to 08:00 (PRE_Time_0808). The first four meteorological factors were
utilized to simulate the autumn phenology model of pollen, predicting various pollen
sDOY and eDOY. All meteorological factors served as training datasets for the RF
algorithm to simulate annual pollen production.

For land use data, this study employed the Community Land Model 4 (CLM4)



dataset (Oleson et al., 2010), which includes 25 plant functional types such as
needleleaf forests, broadleaf forests, shrubs, grasses (C3 and C4), and crops, with a
spatial resolution of 0.05°. As Artemisia and Chenopod primarily fall under the C3
plant category (Yorimitsu et al., 2019; Septembre-Malaterre et al., 2020; Qiao et al.,
2023), the simulation of pollen utilization for Artemisia and Chenopod used plant
functional C3 grass, while the TotalPC simulation incorporated both C3 and C4
grasses. The distribution of these two plant functional types in Beijing is illustrated in
Fig. 2.

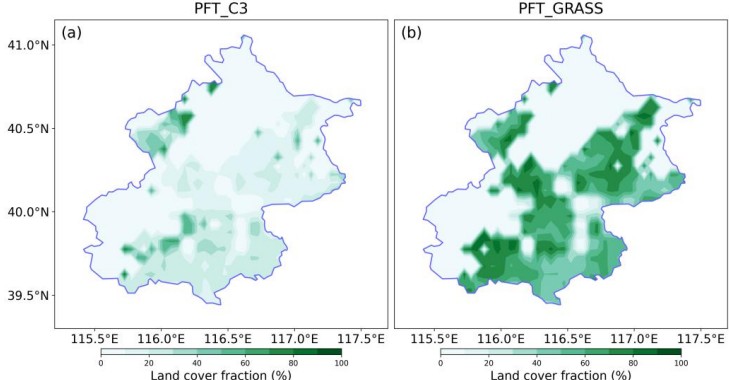


Fig. 2. The distribution of plant functional type C3 (a) and GRASS (b) in Beijing area

**3. Results and Discussion**
*3.1 Pollen Phenology Simulation*

In this study, we analyzed the phenological changes of three types of

pollen—Artemisia, Chenopod, and TotalPC—during the autumn season based on
three different autumn pollen phenology calculation methods ($Rs_1$, $Rs_2$, and $Rs_{sig}$).
Specifically, we examined the seasonal phenological simulations of these pollen
concentrations under three different temperature conditions (TEM_Avg, TEM_Max,
and TEM_Min) (Mo et al., 2023), with a primary focus on sDOY and eDOY.
Additionally, the annual pollen production ($P_{annual}$) was simulated using the RF
algorithm.
3.1.1 Simulation of sDOY and eDOY based on autumn phenology model

Table 2 presents the statistical indicators for simulating the phenology of



Artemisia using different phenological methods and temperature conditions. For
simulating the sDOY for Artemisia, the $Rs_1$, $Rs_2$, and $Rs_{sig}$ methods demonstrated high
accuracy when TEM_Avg and TEM_Min were employed as temperature conditions.
The R values for both the training and testing sets exceeded 0.45, with some R values
in the testing set surpassing 0.7, and the RMSE values were relatively low. This
indicates that these three methods effectively capture the phenological characteristics
of Artemisia at the onset of autumn. Notably, the $Rs_{sig}$ method, when using TEM_Avg
as the condition, achieved R values of 0.53 and 0.80 for the training and testing sets,
respectively, with RMSE values of 6.61 and 4.86, showing the best simulation
performance. However, when TEM_Max was used as the temperature condition, the
simulation performance of all three methods declined. The R value of the $Rs_1$ method
fell below 0.2, and the RMSE values were high, exceeding 8 days. Comparatively, the
$Rs_{sig}$ method performed slightly better but still yielded inferior results compared to
TEM_Avg and TEM_Min, indicating lower model stability when predicting Artemisia
sDOY with TEM_Max. For the simulation of Artemisia eDOY, the performance of the
three methods was relatively close, with R values in the training and testing sets
generally ranging from 0.3 to 0.5, and similar RMSE values. Among them, the $Rs_1$
method performed better when TEM_Min and TEM_Avg were used as temperature
conditions, with R values of 0.66 and 0.51 in the testing set and RMSE values of 3.32
days and 3.9 days, respectively. Compared to the $Rs_1$ method, the $Rs_2$ and $Rs_{sig}$
methods were relatively weaker in predicting eDOY, indicating that the $Rs_1$ method
better captures the phenological trends of Artemisia at the end of autumn. Additionally,
when comparing the simulation results of sDOY and eDOY, sDOY generally had
higher R values, but eDOY had lower overall RMSE values.

The statistical indicators for simulating the phenology of Chenopod under

different phenological methods and temperature conditions are shown in Table 3. For
the simulation of the sDOY for Chenopod, the $Rs_1$ and $Rs_2$ methods demonstrated
high accuracy when using TEM_Min and TEM_Avg as temperature conditions. The R
values for both the training and testing sets were around 0.5, and the RMSE values
were relatively low. It is clear that using TEM_Avg as the temperature condition



yields higher R values and lower RMSE (in the case of the $Rs_1$ method) compared to
TEM_Min, indicating that these two methods effectively capture the phenological
changes of Chenopod at the onset of autumn when using TEM_Avg as the
temperature condition. However, when TEM_Max was used as the temperature
condition, the simulation performance of all three methods declined, particularly for
$Rs_1$, which had an R value of -0.1 and an RMSE greater than 9 days in the testing set.
The $Rs_{sig}$ method, when using TEM_Avg, achieved an R value of 0.51 in the training
set but only 0.28 in the testing set, with a high RMSE of 5.32, indicating poor model
stability in this scenario. In contrast to TotalPC and Artemisia, the simulation of the
eDOY for Chenopod was not satisfactory for any of the three methods. The R values
for both the training and testing sets were all below 0.42. Particularly when using
TEM_Max as the temperature condition, the simulation performance of all three
methods was poor, with the testing set R value reaching only 0.1. This indicates that
the models have limited ability to capture the end of the autumn season for Chenopod.
Table 4 shows the phenological simulation statistical indicators of TotalPC under
different phenological methods and temperature conditions. From the data in the table,
it can be seen that for the simulation of the sDOY of TotalPC, all three phenological
methods ($Rs_1$, $Rs_2$, and $Rs_{sig}$) performed with high accuracy (R > 0.5) and relatively
low RMSE when using TEM_Min. This indicates that these three methods, when
using TEM_Min, can effectively capture the trend of the sDOY of TotalPC during the
autumn season. Meanwhile, the $Rs_1$ method also showed good simulation
performance when using TEM_Avg as the temperature condition, with R reaching
0.54 for both the training and testing sets. The $Rs_{sig}$ method, using TEM_Avg, had
good simulation performance in the training set, but the R in the testing set only
reached 0.38. Compared to TEM_Min and TEM_Avg, the $Rs_2$ and $Rs_{sig}$ methods
showed slightly inferior simulation performance when using TEM_Max as the
temperature condition. Surprisingly, the $Rs_1$ method's simulation of the sDOY showed
a negative correlation when using TEM_Max, indicating the worst performance. For
the simulation of the eDOY of TotalPC, the overall simulation performance was
worse in terms of R compared to sDOY, but the RMSE values were generally better.



Specifically, using TEM_Avg as the temperature condition, the $Rs_2$ and $Rs_{sig}$ methods
showed relatively good simulation performance and lower RMSE. However, the $Rs_2$
method performed much worse on the testing set compared to the training set, with
the R on the testing set being only 0.32.
Overall, different pollen types exhibit varying sensitivity to different
phenological models and temperature conditions. TEM_Avg is generally the best
temperature condition for predicting the sDOY of the three pollen types, providing
higher R values and lower RMSE. This suggests that TEM_Avg can effectively
predict the start of the autumn pollen season. At the same time, TEM_Min also
performs well in predicting the sDOY of TotalPC and Artemisia, whereas TEM_Max
generally shows the poorest prediction performance. For predicting eDOY, different
pollen types show different sensitivities to temperature conditions, but overall, the
models perform worse for eDOY compared to sDOY, especially in the simulation of
Chenopod.
Table2 Statistical indicators of Artemisia phenology under different phenological methods and

temperature conditions

| Artemisia | | $Rs_1(R)$ | | $Rs_2(R)$ | | $Rs_{sig}(R)$ | | $Rs_1(RMSE)$ | | $Rs_2(RMSE)$ | | $Rs_{sig}(RMSE)$ | |
|---|---|---|---|---|---|---|---|---|---|---|---|---|---|
| | | Train | Test | Train | Test | Train | Test | Train | Test | Train | Test | Train | Test |
| sDOY | TEM_Min | 0.47 | 0.66[#] | 0.52[*] | 0.77[#] | 0.45 | 0.59[#] | 6.61 | 5.93 | 6.29 | 4.99 | 6.63 | 6.57 |
| | TEM_Avg | 0.45 | 0.63[#] | 0.50 | 0.71[#] | **0.53[*]** | **0.80[#]** | 6.67 | 6.18 | 6.78 | 5.44 | 6.61 | 4.86 |
| | TEM_Max | 0.16 | 0.17 | 0.44 | 0.47 | 0.45 | 0.58[#] | 8.87 | 9.58 | 8.21 | 7.51 | 6.52 | 6.32 |
| eDOY | TEM_Min | **0.38** | **0.66[#]** | 0.38 | 0.44 | 0.36 | 0.37 | 4.19 | 3.32 | 4.19 | 3.97 | 4.02 | 4.07 |
| | TEM_Avg | 0.46 | 0.51[*] | 0.38 | 0.29 | 0.44 | 0.44 | 3.92 | 3.9 | 4.16 | 4.23 | 3.85 | 4.07 |
| | TEM_Max | 0.31 | 0.43 | 0.05 | 0.07 | 0.33 | 0.27 | 5.59 | 4.65 | 6.84 | 6.47 | 3.98 | 4.32 |

Table3 Statistical indicators of Chenopod phenology under different phenological methods and

temperature conditions

| Chenopod | | $Rs_1(R)$ | | $Rs_2(R)$ | | $Rs_{sig}(R)$ | | $Rs_1(RMSE)$ | | $Rs_2(RMSE)$ | | $Rs_{sig}(RMSE)$ | |
|---|---|---|---|---|---|---|---|---|---|---|---|---|---|
| | | Train | Test | Train | Test | Train | Test | Train | Test | Train | Test | Train | Test |
| ○ | TEM_Min | 0.42 | 0.44 | 0.59[#] | 0.36 | 0.47 | 0.36 | 4.68 | 5.09 | 4.12 | 5.28 | 4.38 | 5.25 |




|  |  | Rs₁(R) | | Rs₂(R) | | Rs_sig(R) | | Rs₁(RMSE) | | Rs₂(RMSE) | | Rs_sig(RMSE) | |
|---|---|---|---|---|---|---|---|---|---|---|---|---|---|
|  | TEM_Avg | 0.55* | 0.47 | **0.63#** | **0.33** | 0.51 | 0.28 | 4.13 | 5.37 | 4.49 | 5.42 | 4.12 | 5.32 |
|  | TEM_Max | 0.31 | -0.1 | 0.43 | 0.28 | 0.42 | 0.18 | 7.84 | 9.13 | 5.66 | 5.9 | 5.55 | 6.06 |
|  | TEM_Min | **0.42** | **0.25** | 0.17 | 0.2 | 0.26 | 0.11 | 4.23 | 4.94 | 4.31 | 4.71 | 4.15 | 4.75 |
| eDOY | TEM_Avg | 0.37 | 0.1 | 0.39 | 0.23 | 0.34 | 0.33 | 3.98 | 5.29 | 4.09 | 4.84 | 4.16 | 4.65 |
|  | TEM_Max | 0.23 | -0.0 | 0.27 | -0.1 | 0.13 | 0.11 | 5.57 | 6.87 | 5.53 | 7.09 | 6.31 | 7.14 |

Table4 Statistical indicators of TotalPC phenology under different phenological methods and temperature conditions

| TotalPC | | Rs₁(R) | | Rs₂(R) | | Rs_sig(R) | | Rs₁(RMSE) | | Rs₂(RMSE) | | Rs_sig(RMSE) | |
|---|---|---|---|---|---|---|---|---|---|---|---|---|---|
|  |  | Train | Test | Train | Test | Train | Test | Train | Test | Train | Test | Train | Test |
|  | TEM_Min | 0.52* | 0.53# | **0.59#** | **0.56#** | 0.58# | 0.55# | 5.84 | 5.32 | 5.51 | 5.32 | 5.61 | 5.6 |
| sDOY | TEM_Avg | 0.54# | 0.54# | 0.08 | nan | 0.59# | 0.45 | 5.89 | 5.21 | 6.75 | 6.27 | 5.71 | 5.62 |
|  | TEM_Max | -0.21 | -0.19 | 0.51* | 0.48* | 0.52* | 0.4 | 9.04 | 9.2 | 7.66 | 6.45 | 5.83 | 6.1 |
|  | TEM_Min | 0.41 | 0.21 | 0.35 | 0.24 | 0.5* | 0.36 | 4.76 | 4.47 | 4.9 | 4 | 4.75 | 3.41 |
| eDOY | TEM_Avg | 0.51* | 0.18 | 0.63# | 0.32 | **0.5*** | **0.49*** | 4.47 | 4.83 | 4.4 | 3.95 | 4.63 | 3.11 |
|  | TEM_Max | 0.44 | 0.4 | 0.18 | 0.2 | 0.39 | 0.29 | 6.41 | 6.47 | 7.7 | 6.56 | 4.78 | 3.72 |

Note: Bold represents the best model performance, # Indicates significance levels at P < 0.001, * Indicates significance levels at P < 0.005

Based on the above discussion, we selected the most suitable phenological and temperature conditions for the three types of pollen (bold parts in Table 2-4), simulated their sDOY and eDOY, and generated line and scatter plots (Fig. 3). According to the line plots in Fig. 3 (top), the predicted results for Artemisia are the closest to the actual observed results. The predictions for TotalPC follow, while the predictions for Chenopod show some deviation, particularly in eDOY, indicating the need for a more suitable phenological model to accurately simulate the phenology of Chenopod. The scatter plots in Fig. 3 (bottom) illustrate that for sDOY predictions, Artemisia exhibited the strongest correlation between predicted and observed pollen phenology, with an R value of 0.69 and an RMSE of 5.77 days. In contrast, Chenopod had the lowest correlation, with an R value of 0.49 and an RMSE of 4.98 days. It can also be observed that higher R values are associated with higher overall RMSE,



possibly due to the models being more sensitive to noise or outliers in the data, which
increases the overall error. For high-correlation predictions like those for Artemisia,
the model may be more affected by random fluctuations in the data, leading to
increased error. Additionally, different pollen types may exhibit varying
characteristics or response patterns in phenological models, resulting in a non-linear
or inconsistent relationship between correlation and error. For eDOY predictions, the
correlation between predicted and observed is highest for Artemisia, with an R value
of 0.53 and an RMSE of 3.77 days. Chenopod has the lowest correlation for eDOY
predictions, with an R value of only 0.26 and an RMSE of 4.57 days. The poorer
performance in simulating eDOY for Chenopod may be due to lower data quality
compared to Artemisia and TotalPC, as well as the smallest sample size, resulting in
insufficient information and samples for the model to learn and predict accurately.
Additionally, Table 5 shows the proportion of simulations with errors less than 5
days and 3 days for sDOY and eDOY across the three pollen types. It can be seen that
the proportion of eDOY simulations with errors less than 5 days and 3 days is higher
than that for sDOY, indicating that eDOY simulations generally have better accuracy
in terms of error. Specifically, for Chenopod eDOY simulations, although the R value
is poor, 76.79 % of simulations have errors less than 5 days, and 55.36 % have errors
less than 3 days, meaning that more than half of the eDOY simulations have errors
within 3 days. This performance is comparable to the other two pollen types (64.41 %
and 68.12 %, respectively). Compared to Mo et al. (2023), which simulated the spring
season start pollen season (SPS) using 17 phenological models, this study has slightly
lower R values but much lower RMSE (around 11 days in their study). Li et al. (2022)
used satellite data to simulate the SPS for Birch, Oak, and Poplar, achieving RMSE
values between 4.26 and 8.77 days. Furthermore, this study's process-based
phenological models for sDOY and eDOY show smaller errors and higher
correlations compared to empirical linear models based solely on temperature used by
Wozniak and Steiner (2017) and Zhang and Steiner (2022).
Therefore, from an error analysis perspective, the simulation performance of
Chenopod eDOY maintains a relatively low error while also demonstrating some





stability, indicating that the autumn phenological model can accurately capture the
seasonal variation trend of Chenopod. This makes the simulation results reliable.
Overall, the autumn phenological models provide good simulation performance for
the phenology of the three pollen types, laying a solid foundation for further analysis
of pollen temporal characteristics.

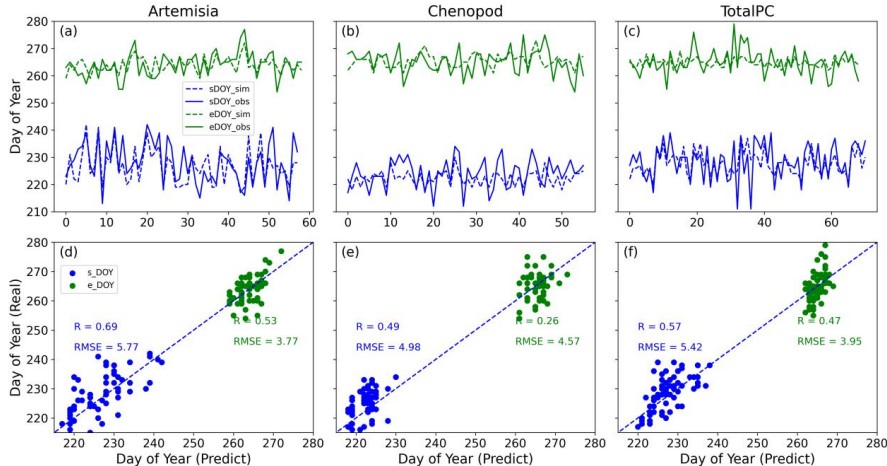


Fig. 3. Comparison of pollen sDOY and eDOY in autumn phenology: simulation vs. observation.
Line plots of three different pollen sDOY and eDOY (a-c) and scatter plot comparison of the same
(d-f). Specific comparisons for Artemisia (a, d), Chenopod (b, e), and TotalPC (c, f).
Table5 Statistics on the proportion of errors between simulation and observation of three different
types of pollen sDOY and eDOY within 5 and 3 days

|  | DOY | Artemisia (%) | Chenopod (%) | TotalPC (%) |
|---|---|---|---|---|
| <5D | sDOY | 68.97 | 73.21 | 71.83 |
|  | eDOY | 86.44 | 76.79 | 82.61 |
| <3D | sDOY | 48.28 | 44.64 | 53.52 |
|  | eDOY | 64.41 | 55.36 | 68.12 |

Based on the temperature and SSH observational station data from the Beijing
area, we interpolated the station data into a grid dataset with a horizontal resolution of
0.1°. Using the selected autumn phenological models, we then performed gridded



simulations of the sDOY and eDOY for three pollen types. This approach enabled us
to map the regional distribution of autumn pollen sDOY and eDOY in Beijing from
2006 to 2020, thereby laying the groundwork for further simulations of autumn pollen
emissions potential.
3.1.2 Simulation of annual pollen production based on RF
The simulation of annual pollen production ($P_{annual}$, referring to the cumulative
pollen concentration during each autumn pollen season) was conducted using the RF
algorithm. The training data comprised all station-observed pollen data from Table 1
and the corresponding meteorological observation data from Sect. 2.2.2. Four-fifths of
the station data were randomly selected as the training set to train the RF algorithm,
while the remaining one-fifth was used as the test set to validate the accuracy of the
RF's $P_{annual}$ simulation. Fig. 4 presents the scatter plots of observed versus simulated
$P_{annual}$ for three different pollen types (Artemisia, Chenopod, and TotalPC) based on
the RF in the test set. The R between simulated and observed values for the three
pollen types were all above 0.5, with Chenopod reaching 0.65. The calculated RMSE
was around $0.2 \times 10^6$ Grains m$^{-2}$ year$^{-1}$ (with TotalPC having an RMSE of $2.12 \times 10^6$
Grains m$^{-2}$ year$^{-1}$). This indicates that the prediction performance of the RF varies
among different pollen types, with the best performance for Chenopod and the poorest
for TotalPC annual production. Compared to the temperature-based empirical linear
models for $P_{annual}$ by Zhang and Steiner (2022), the machine learning algorithm-based
simulations in this study have smaller errors and higher correlations. Overall, the RF
effectively simulates $P_{annual}$.
Based on meteorological observation data from stations in and around Beijing,
the station data were interpolated into a gridded dataset with a horizontal resolution of
0.1°. Subsequently, all station data for each pollen type were used as the training set,
with 12 stations in the gridded dataset cyclically selected as the test set for gridded
simulations. This ultimately resulted in the spatial distribution of $P_{annual}$ in Beijing
from 2006 to 2020, laying the foundation for further simulation of autumn pollen
emissions potential.



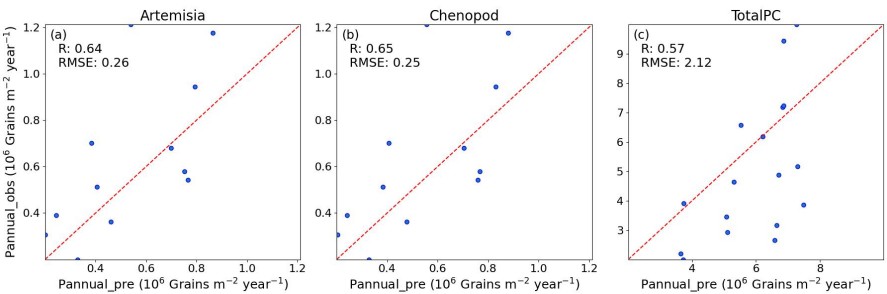

Fig. 4. Scatter plots of simulated and observed annual pollen ($P_{annual}$) based on RF. Comparisons for Artemisia (a), Chenopod (b), and TotalPC (c).

*3.2 Simulation of Pollen Emissions in Beijing Area*

Based on the simulation results of autumn pollen phenology (sDOY, eDOY, and $P_{annual}$) from Sect. 3.1 and the pollen emissions potential parameterization method from Sect. 2.1.1, this study calculated the pollen emissions potential in the Beijing area. Fig. 5-7 present a comparison between the observed and simulated average site values of Artemisia, Chenopod, and TotalPC in Beijing from 2006 to 2020. In these figures, blue dots represent the actual daily observed pollen counts, and red lines represent the simulated pollen emissions. To assess the consistency between the simulated and observed data, we calculated R and RMSE. As illustrated in the figures, the simulated data closely match the actual observations in most years, with correlation coefficients around 0.9. Specifically, the Artemisia emissions in 2010, Chenopod emissions in 2016, and TotalPC emissions in 2007, 2009, 2018, and 2019 show R values as high as 0.98 and relatively low RMSE levels, demonstrating the high accuracy of this study in simulating pollen emissions potential.

Additionally, the simulation results for sDOY and eDOY were also satisfactory, though there were slight advances in the start of the pollen season in certain years, such as 2017 and 2018 for Artemisia and Chenopod. While the peak pollen emissions simulations were highly accurate in most years, there were instances of overestimation and underestimation in some years. For example, the peak emissions of Artemisia in 2008, 2009, and 2020, Chenopod in 2007, and TotalPC in 2013 and 2020 were significantly underestimated. Conversely, the peak simulations of TotalPC in 2011 and 2012 were slightly overestimated. This indicates that, despite the high



accuracy of the annual pollen production simulations based on the RF, there is still
room for improvement
Overall, this study achieved significant results in simulating pollen emissions,
demonstrating the potential application of autumn phenological models and the RF
algorithm in simulating pollen emissions. However, to further enhance the accuracy of
these simulations, future research needs to investigate and address the instances of
overestimation and underestimation in greater detail.

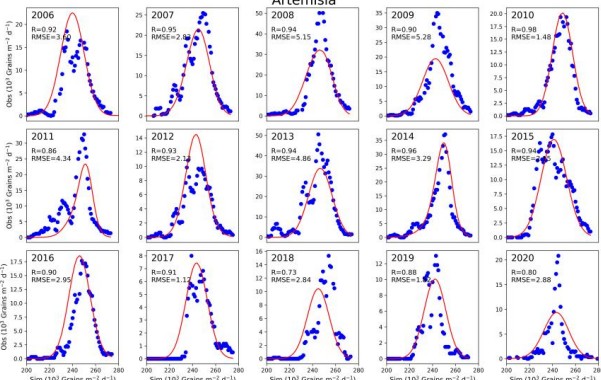


Fig. 5. Time series of observation and simulation of average Artemisia emissions at stations in
Beijing from 2006 to 2020. The red solid line represents the simulation of pollen emissions model,

while blue dots depict observations

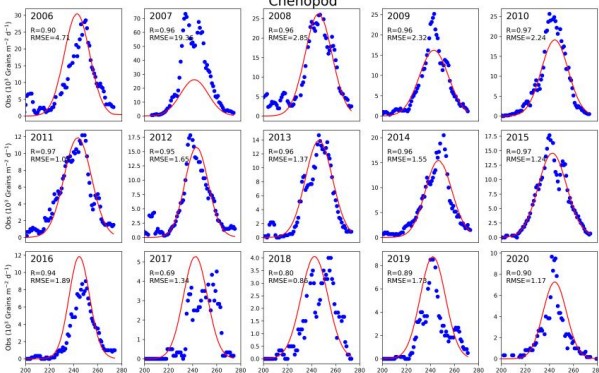


Fig. 6. Time series of observation and simulation of average Chenopod emissions at stations in
Beijing from 2006 to 2020. The red solid line represents the simulation of pollen emissions model,



while blue dots depict observations

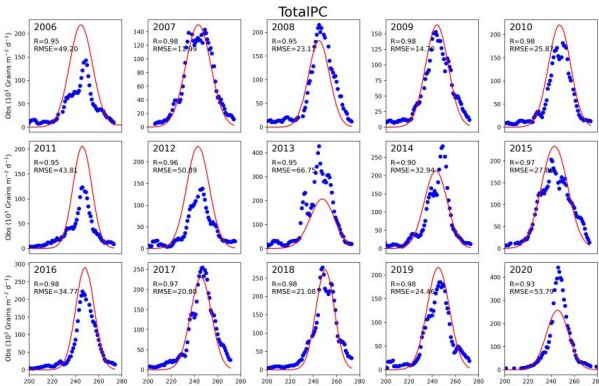


Fig. 7. Time series of observation and simulation of average TotalPC emissions at stations in
Beijing from 2006 to 2020. The red solid line represents the simulation of pollen emissions model,
while blue dots depict observations
To further investigate the spatial distribution of annual pollen production, we
simulated the spatial distribution of annual Artemisia, Chenopod, and TotalPC
production in Beijing from 2006 to 2020 (Fig. 8-10). The results reveal significant
spatial and temporal variations in annual pollen production. Spatially, Artemisia
production is predominantly concentrated in the southeastern, northeastern, and
certain northwestern regions of Beijing, with occasional occurrences in the central
urban area during specific years (2008 and 2013). Chenopod production is highest in
the southern part of Beijing and lowest in the northern parts and surrounding areas.
Notably, from 2006 to 2008, the southern region exhibited high concentrations of
Chenopod production. TotalPC is mainly distributed in the southeastern plains of
Beijing, forming a strip-like pattern, while lower production is observed in the
northwestern mountainous areas, indicating a possible influence of geographical
location on TotalPC distribution. Temporally, the annual production of these three
pollen types demonstrates distinct interannual variations. Artemisia shows little
change in both distribution area and production concentration over time. In contrast,
Chenopod and TotalPC exhibit a general declining trend, reaching their lowest levels
between 2016 and 2018, which may be attributed to recent climatic changes,





vegetation shifts, and human activities in the Beijing area.

The simulation results for annual pollen production of Artemisia, Chenopod, and

TotalPC in Beijing from 2006 to 2020, based on autumn phenology and the RF pollen
emissions model, indicate pronounced spatial differences and temporal variation
characteristics. Analyzing the spatial distribution and temporal variation of annual
pollen production in Beijing enhances our understanding of the spatiotemporal
patterns of pollen in the region, providing crucial insights for the control and
mitigation of pollen allergies.

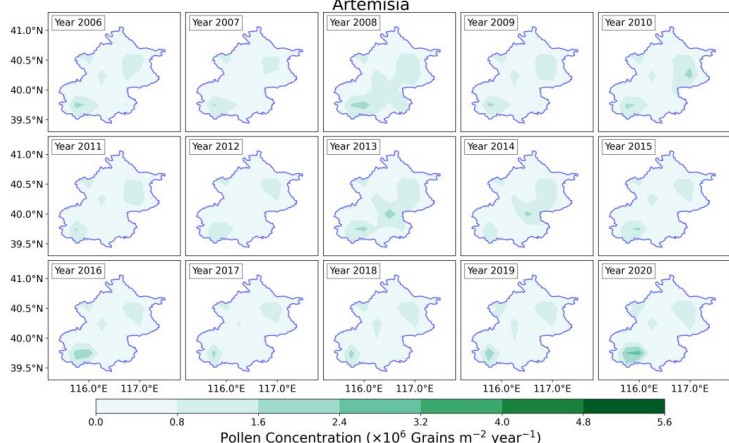


Fig. 8. Distribution of Artemisia in Beijing from 2006 to 2020 based on pollen emissions model

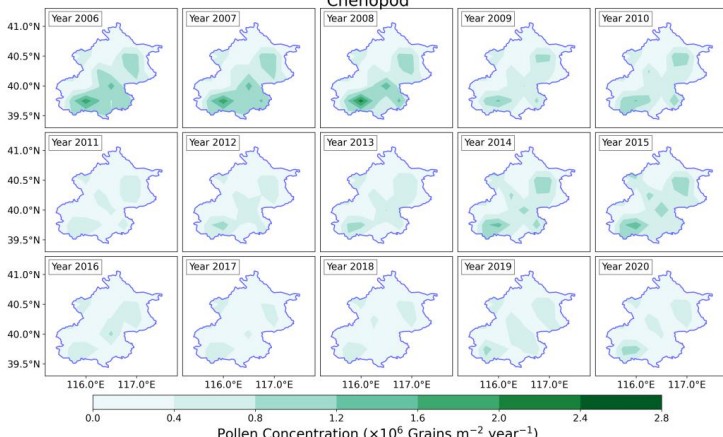


Fig. 9. Distribution of Chenopod in Beijing from 2006 to 2020 based on pollen emissions model



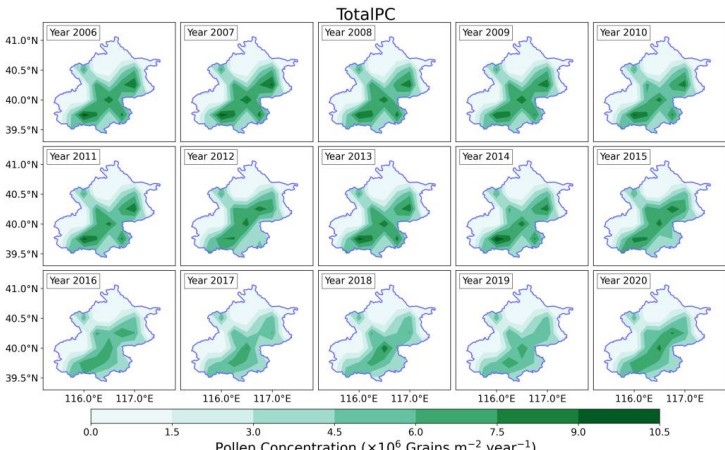


Fig. 10. Distribution of TotalPC in Beijing from 2006 to 2020 based on pollen emissions model

To more intuitively reflect the temporal variation trends in the annual production

of three types of pollen, we further analyzed the interannual variation of the regional
average cumulative concentration of these pollen types during the autumn pollen
season in Beijing from 2006 to 2020 (Fig. 11). The annual production of Artemisia,
Chenopod, and TotalPC in Beijing averages between 0.8-1.6, 0.5-1.4, and 6.5-9 grains
m-2 year-1, respectively. The annual production of Artemisia and Chenopod are
notably similar. Over time, the regional annual production of these pollen types in
Beijing exhibits significant fluctuations. Nonetheless, Artemisia remains relatively
stable, whereas Chenopod and TotalPC production demonstrate a discernible
declining trend, particularly in TotalPC. The annual production of all three pollen
types reached a local nadir in 2012. Following a surge in 2013, production steadily
declined from 2014 to 2017, reaching the lowest levels observed in nearly 15 years
(with TotalPC being the lowest in 2018). Subsequently, from 2018 to 2020, an
increasing trend was observed. Overall, the annual pollen production in Beijing
appears to follow a minor cyclical pattern, intimately linked to the impacts of climate
change.

To further explore the meteorological factors influencing average annual pollen

production in Beijing, we selected six meteorological variables during the autumn
pollen season from 2006 to 2020 for temporal and regional average calculations.





These factors include maximum temperature (TEM_Max), average temperature
(TEM_Avg), minimum temperature (TEM_Min), average relative humidity
(RHU_Avg), sunshine hours (SSH), and precipitation time (PRE_Time_0808). The
annual variations of these meteorological factors were analyzed, and their correlations
with annual pollen production variations were calculated (Fig. 12).
The trends in annual variations of each meteorological factor and the calculated
correlations reveal that for Artemisia, TEM_Min and RHU_Avg have a significant
positive correlation with its production, especially RHU_Avg, which shows a
correlation of 0.79. This indicates that an increase in relative humidity promotes
Artemisia production. Conversely, SSH has a correlation of -0.8 with Artemisia,
indicating that longer sunshine hours inhibit its production. Meanwhile, TEM_Avg
and PRE_Time_0808 have minor promoting effects on Artemisia production, while
TEM_Max has a slight inhibitory effect. For Chenopod, TEM_Min is the most
significant promoting factor, while SSH has an inhibitory effect, although its negative
correlation is lower than that for Artemisia, indicating a limited inhibitory effect on
Chenopod production. For TotalPC, similar to Artemisia, increases in TEM_Min and
RHU_Avg promote production, while increases in SSH and TEM_Max inhibit
production. Notably, the three types of pollen reached local minimum concentrations
in 2012, 2017, and 2018, when TEM_Min and SSH respectively reached local
minimum and maximum values, further demonstrating the promoting effect of
TEM_Min and the inhibitory effect of SSH on annual average pollen concentration.
Rahman et al. (2020) and Lei et al. (2023) indicated that temperature is the main
factor affecting the interannual variation of pollen and is positively correlated with
pollen production. Our findings are largely consistent with these conclusions,
although they did not consider the effect of SSH on interannual changes in pollen
concentration. In summary, the annual production of pollen in Beijing is significantly
influenced by meteorological conditions, particularly temperature, relative humidity,
and sunshine hours. Different meteorological factors exhibit distinct promoting and
inhibiting effects on pollen production.



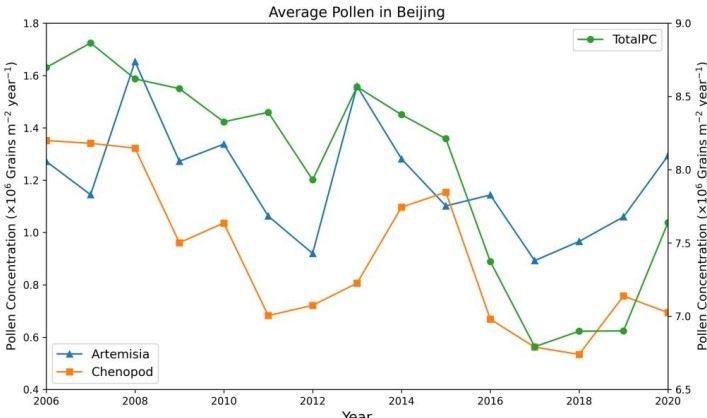


Fig. 11. Time series variation chart of regional average annual production of three types of pollen

in Beijing from 2006 to 2020

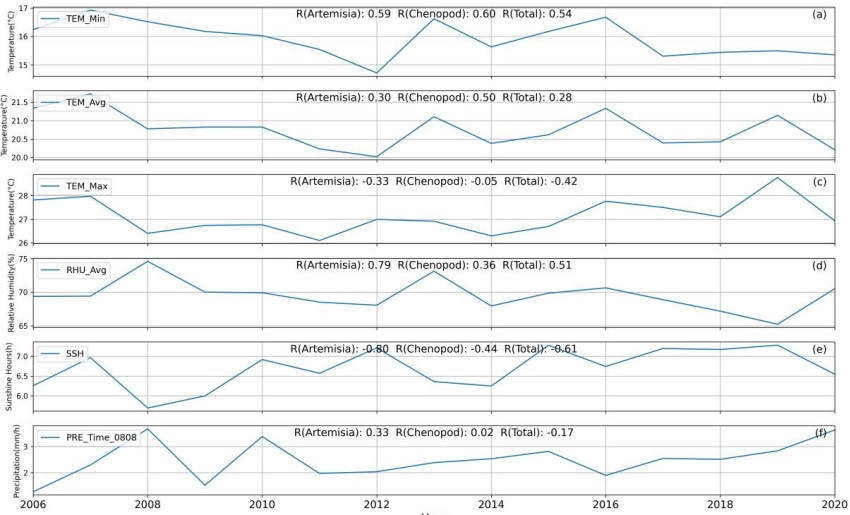


Fig. 12. Time series variation chart of average values of different meteorological factors in Beijing

from 2006 to 2020. (The correlation coefficient between the average meteorological factors and

the regional average annual production of three types of pollen is calculated in the figure)

Fig. 13-15 illustrate the spatial distribution of the average concentrations of Artemisia, Chenopod, and TotalPC during the autumn pollen season in Beijing from 2006 to 2020. During this period, the concentration of all three pollen types initially increases and then decreases. The pollen season begins around August 10 each year and concludes around September 25. The peak concentrations for Artemisia and



Chenopod pollen occur around August 30, while the peak concentration for TotalPC is
observed around September 5. The entire pollen season lasts approximately 45 days.
Regarding the average pollen concentration distribution, Artemisia is primarily
concentrated in the southwest, northeast, and parts of the northwest of Beijing, with
lower concentrations in the southeast. In contrast, Chenopod and TotalPC are mainly
distributed in the southeastern plains. The maximum average concentrations for
Artemisia, Chenopod, and TotalPC reach $81.1\times10^3$ Grains $m^{-2}$ $d^{-1}$, $42.0\times10^3$ Grains
$m^{-2}$ $d^{-1}$, and $351.8\times10^3$ Grains $m^{-2}$ $d^{-1}$, respectively.

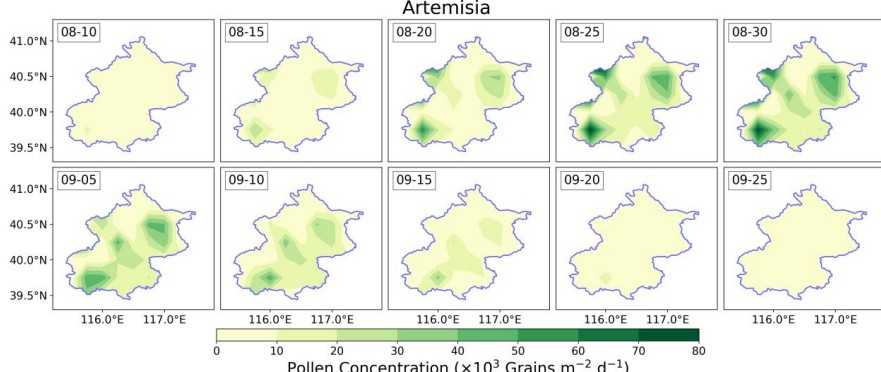


Fig. 13. Temporal and spatial distribution of Artemisia in Beijing (average from 2006 to 2020)

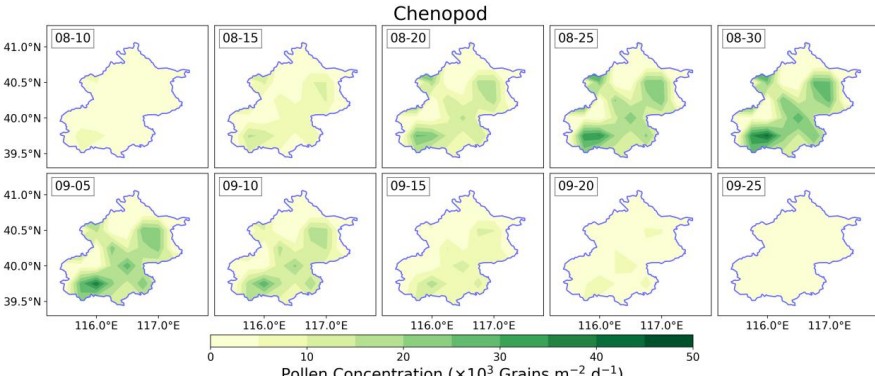


Fig. 14. Temporal and spatial distribution of Chenopod in Beijing (average from 2006 to 2020)



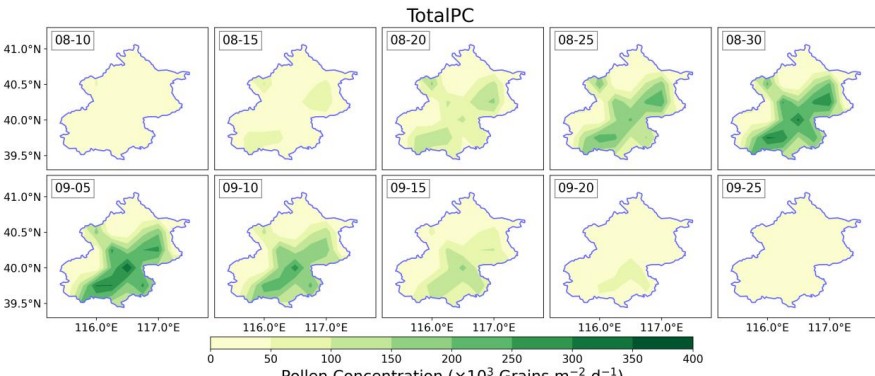


Fig. 15. Temporal and spatial distribution of TotalPC in Beijing (average from 2006 to 2020)


*3.3  Simulation of Pollen Emissions in Regional Climate Models*

To evaluate the pollen emissions model based on autumn pollen phenology and

RF, this study integrates the offline calculated pollen emissions into the regional
climate model RegCM. By comparing the simulated atmospheric pollen
concentrations with data from ground-based pollen monitoring stations, we assess the
performance of this pollen emissions potential model.
3.3.1 Implementation of pollen emissions in Regional Climate Model (RegCM)

RegCM is the pioneering regional climate model system used for climate

downscaling, originating in the late 1980s and early 1990s at the National Center for
Atmospheric Research (NCAR) in the USA. It has since undergone several
development iterations and is currently maintained at the International Centre for
Theoretical Physics (ICTP) in Italy. This open-source system is widely utilized by
numerous research teams, forming an extensive network for regional climate research.
The model can be applied globally and is evolving into a fully coupled regional earth
system model, incorporating ocean, lake, aerosol, desert dust, chemistry, hydrology,
and land surface processes. The version used in this study is RegCM4.7.1.

In this model, a pollen emissions model based on phenology and RF calculates

the emission potential of different types of pollen offline, and then incorporated into
the RegCM model. The calculation of pollen concentration in this model follows the
method of Sofiev et al. (2013), with the formula as follows:






$$E_{pollen,i}(t) = E_i(t) \bullet u_{star} \bullet ce \bullet f_w \bullet f_r \bullet f_h / htc \qquad (8)$$

$$f_w = 1.5 - \exp(-(u_{10} + u_{conv})/5)$$


$$f_r = \begin{cases} 1, pr < pr_{low} \\ \dfrac{pr_{high} - pr}{pr_{high} - pr_{low}}, pr_{low} < pr < pr_{high} \\ 0, pr > pr_{high} \end{cases} \qquad (9)$$

$$f_h = \begin{cases} 1, rh < rh_{low} \\ \dfrac{rh_{high} - rh}{rh_{high} - rh_{low}}, rh_{low} < rh < rh_{high} \\ 0, rh > rh_{high} \end{cases}$$

Where $f_w$, $f_r$ and $f_h$ represent the wind, precipitation, and RH factors,
respectively, influencing pollen emissions concentration. $u_{star}$ is surface friction
velocity, $ce$ is flowering factor, and $htc$ is canopy height. $f_w$ is exponentially related to
the 10m wind speed $u_{10}$ and vertical turbulent wind speed $u_{conv}$. $pr$ and $rh$ represent
precipitation and RH. When precipitation is below the threshold $pr_{low}$, the
precipitation factor is 1. When precipitation exceeds the threshold $pr_{high}$, the factor is 0.
When precipitation is between these thresholds, the factor is calculated as the ratio of
the difference between the high threshold and precipitation to the difference between
the thresholds, with default values $pr_{low}=10^{-5}$ mm and $pr_{high}=0.5$ mm. Similarly, the
RH factor is related to RH and its thresholds, with default values $rh_{low}=50$ % and
$rh_{high}=80$ %. These factors explain the impact of wind, precipitation, and humidity on
pollen emissions. Given the significant influence of precipitation and RH on pollen
emissions, this study adjusts $pr_{high}$    and $rh_{high}$ values to 1 mm and 90 %, respectively.
Higher thresholds can prevent excessive suppression of pollen emissions under
frequent precipitation and high humidity conditions, thus more accurately simulating
actual pollen concentration changes and better adapting the model to different climatic
conditions.
Moreover, the RegCM includes the pollen tracer transport equation (Solmon et al.
2006), as follows:

$$\frac{\partial \chi}{\partial t} = \overline{V} \cdot \nabla \chi + F_H + F_V + T_C + S - R_{Wls} - R_{Wc} - D_d \qquad (10)$$

Where $\chi$ represents the tracer, $F_H$ and $F_V$ represent horizontal and vertical



diffusion, $T_C$ represents convective transport, $RWls$ and $RWc$ represent large-scale and convective precipitation wet removal processes, respectively, and $Dd$ represents dry removal processes. This transport equation comprehensively considers various physical processes and removal mechanisms of pollen in the atmosphere, allowing the simulation of the entire process from pollen release to atmospheric dispersion and deposition. This provides a foundation for fully describing the spatial distribution and temporal evolution of pollen in the atmosphere, which is crucial for studying pollen dispersion in the air, determining the spatial distribution of pollen concentration, and predicting future changes in pollen concentration.

3.3.2 Evaluation of pollen simulation accuracy in RegCM

Fig. 16-18 depict the time series distribution of the concentrations of three pollen types simulated by the RegCM compared to observed concentrations from 2006 to 2020. The RegCM successfully captures the temporal variation trends of pollen concentrations during the autumn pollen season, generally showing an initial increase followed by a decrease. Daily pollen concentrations fluctuate significantly due to meteorological factors such as temperature, precipitation, and RH, as well as key physical processes like advection, convection, and dry and wet deposition. Overall, the simulated pollen concentrations by the RegCM align well with the observed trends, though some discrepancies remain.

In the simulation of Artemisia (Fig. 16), the sDOY and pollen production vary annually due to meteorological conditions and key physical processes. The annual peak pollen concentrations generally range from $20\text{-}70\times10^3$ Grains $m^{-2}$ $d^{-1}$, while in 2019-2020, observed pollen concentrations exceeded $100\times10^3$ Grains $m^{-2}$ $d^{-1}$, with notable spikes and drops likely due to abrupt meteorological changes or possible issues with the quality of observation data. The RegCM accurately simulates the sDOY and eDOY, displaying a similar frequency to observations. For peak pollen simulations, years such as 2006, 2007, 2010, 2012, 2015, and 2016 show good performance, with R above 0.7, particularly in 2006 and 2016, where R exceeds 0.85 and RMSE is only $4\times10^3$ Grains $m^{-2}$ $d^{-1}$. However, for other years, peak simulations are underestimated to varying degrees. For 2011, although the trend is consistent, the



observed peak is near $50\times10^3$ Grains $m^{-2}$ $d^{-1}$, while the simulated peak is only $12\times10^3$
Grains $m^{-2}$ $d^{-1}$, indicating a significant underestimation. This underestimation is also
noticeable in 2008, 2013, and 2017-2020. In 2019, although the peak concentrations
align, the trend correlation is low (R=0.49), and RMSE is high. The variability in
observation station data quality and quantity could influence these results, with some
years having fewer than six effective stations (minimum of two), impacting the
average and peak values. Box plots (Fig. 19) reveal that Artemisia concentrations in
2019-2020 are more dispersed, suggesting possible anomalies in observation data.
Overall, the R for RegCM simulations ranges from 0.69 to 0.86 (except 2019), with
RMSE between $3.05$-$15.38\times10^3$ Grains $m^{-2}$ $d^{-1}$.

For Chenopod simulations (Fig. 17), the overall performance is similar to

Artemisia. The annual peak concentrations are generally lower, around $20$-$50 \times 10^3$
Grains $m^{-2}$ $d^{-1}$, except for 2007, which reaches $120 \times 10^3$ Grains $m^{-2}$ $d^{-1}$. The years
2006, 2008-2009, 2012-2013, 2015, and 2019 show good simulation performance,
accurately reflecting peak concentrations, particularly in 2016 (R=0.84,
RMSE=$3.11\times10^3$ Grains $m^{-2}$ $d^{-1}$). However, 2007, 2010, 2017-2018, and 2020 exhibit
underestimation, with the exceptionally high observed concentrations in 2007 likely
causing the model's underestimation. Fig. 19 indicates increasing peak concentrations
in recent years (2017-2020) for both Artemisia and Chenopod, with room for
improvement in peak simulations by the RegCM. Despite the lower concentrations
compared to spring pollen, autumn pollen significantly impacts pollen-induced
diseases (pollinosis), prompting more attention and efforts in pollen management,
which contributes to the decreasing trend in monitored pollen concentrations.

TotalPC generally exhibits higher concentration levels compared to Artemisia

and Chenopod (Fig. 18). Annual peak TotalPC can reach $150$-$500\times10^3$ Grains $m^{-2}$ $d^{-1}$,
with the highest observed concentration in 2020 at $745\times10^3$ Grains $m^{-2}$ $d^{-1}$. Due to the
higher quality and completeness of TotalPC monitoring data, the simulation results
are more accurate, with R generally above 0.76 (except 2015, R=0.64). Over 60 % of
the years have R above 0.8, with fewer years showing significant underestimation of
peak concentrations (e.g., 2013). This highlights the critical role of high-quality



pollen monitoring data for accurate simulations. High-quality data enable precise

capturing of pollen concentration trends and peaks, providing robust support for

regional pollen phenology research.

In summary, the RegCM demonstrates high accuracy in simulating the

concentrations of the three pollen types, especially TotalPC. Accurate simulations of

pollen concentrations and peaks enhance the effectiveness of pollen emissions models,

improve health risk warnings, and provide a scientific basis for urban planning and

environmental management.

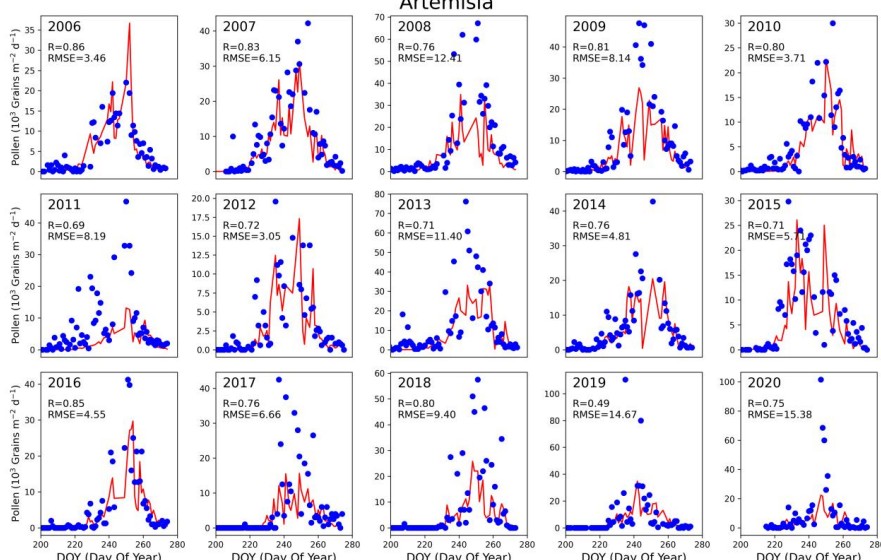

Fig. 16. Time-series distribution of Artemisia under RegCM simulation compared to observations

(averaged across effective pollen monitoring sites). The red solid line represents model

simulations, while blue dots depict observations

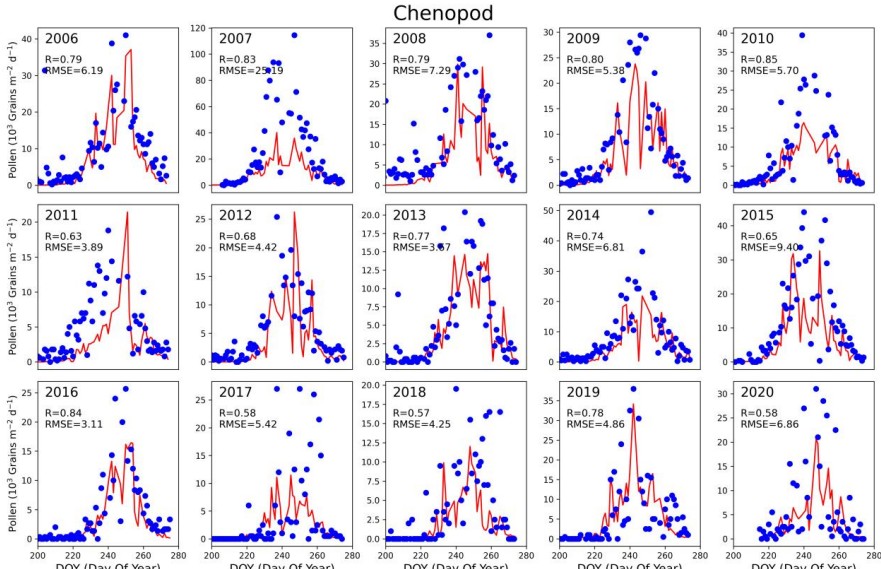

Fig. 17. Time-series distribution of Chenopod under RegCM simulation compared to observations

(averaged across effective pollen monitoring sites). The red solid line represents model

simulations, while blue dots depict observations

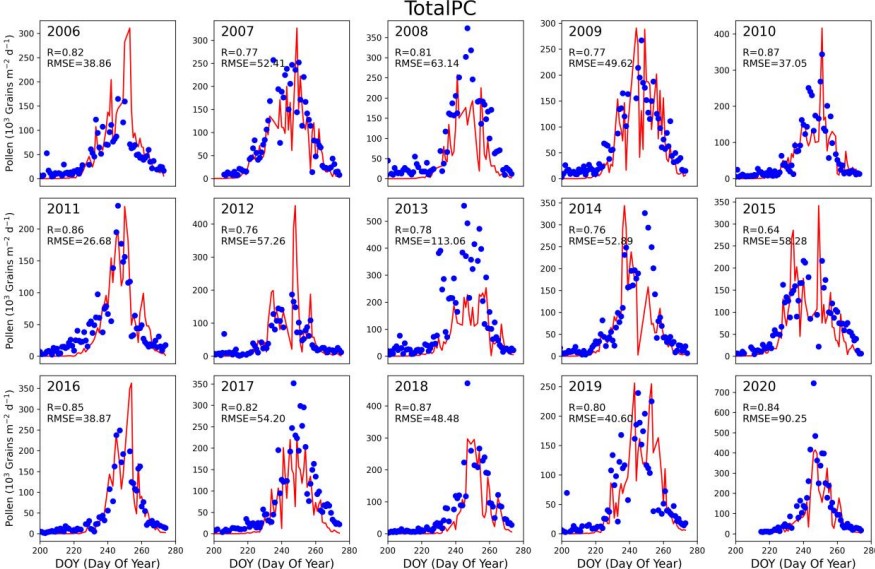

Fig. 18. Time-series distribution of TotalPC under RegCM simulation compared to observations

(averaged across effective pollen monitoring sites). The red solid line represents model

simulations, while blue dots depict observations

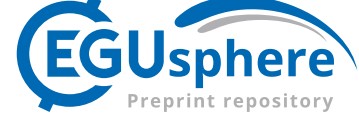

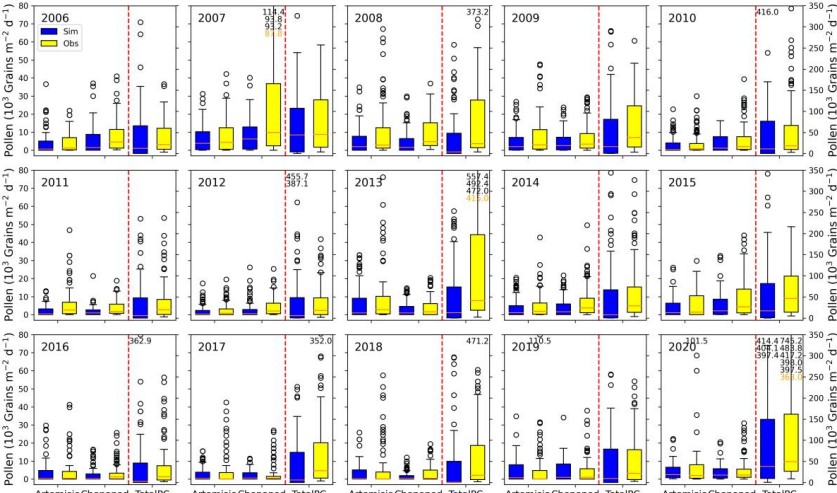

733

Fig. 19. Box plot statistics of pollen concentration under RegCM simulation compared to observed

values. Each subplot features box plots denoted by red dashed lines: on the left side, representing

Artemisia and Chenopod concentrations with values referenced on the left y-axis; on the right side,

depicting TotalPC with values referenced on the right y-axis. In each box plot, from bottom to top,

the box and whiskers indicate the minimum, lower quartile, median, upper quartile, and maximum

values (extending up to 1.5 times the interquartile range, IQR). Black circles denote outliers

exceeding 1.5 times IQR. Orange numbers annotated in the subplot indicate the maximum values

unseen within the box, while black numbers denote unseen outliers

## 4. Conclusion

This study utilized years of autumn pollen concentration data from Beijing,

alongside meteorological and land use data, to develop an autumn pollen emissions

model using autumn phenology and the RF algorithm. We conducted an in-depth

analysis of the spatiotemporal distribution characteristics of Artemisia, Chenopod, and

TotalPC in Beijing and examined their relationships with meteorological factors.

Finally, we validated the accuracy and reliability of the constructed pollen emissions

model using the RegCM. Through a series of simulations and validations, several

significant conclusions and findings were obtained.

(1) Construction of the Pollen Emissions Model: By incorporating phenology



and the RF algorithm, we calculated autumn pollen emissions, thereby avoiding the poor simulation results of sDOY, eDOY, and annual pollen production based solely on temperature linear simulations. The study demonstrates that using a phenology model for sDOY and eDOY simulations captures the temporal variations of pollen release more accurately, effectively reducing simulation errors. The RF algorithm excels in handling multivariate and nonlinear relationships, significantly improving the simulation accuracy of the pollen emissions model. The optimized annual pollen production simulations better reflect seasonal changes in pollen, showcasing the applicability and reliability of the RF algorithm in processing meteorological and environmental data.

(2) Spatiotemporal Distribution Characteristics of Pollen Concentration: The study found significant spatial and temporal variations in pollen concentration in Beijing. The autumn pollen peak occurs between DOY 215-280, with considerable differences in peak times and concentrations among monitoring stations. These differences are closely related to the vegetation types, topographical features, and local climatic conditions around each station. Optimized simulations of pollen concentration data further reveal the spatiotemporal variation patterns of pollen concentrations.

(3) Impact of Meteorological Factors on Annual Pollen Emissions: Meteorological factors significantly influence pollen concentrations. The study reveals that temperature, RH, and SSH are crucial factors affecting annual pollen emissions in Beijing. There is a positive correlation between temperature and RH with annual pollen emissions, while SSH has a negative correlation. The response of different pollen types to meteorological factors varies due to their distinct biological characteristics and ecological environments. This comprehensive analysis provides a scientific basis for predicting future changes in pollen concentrations.

(4) Validation of Pollen Emissions Models Using the RegCM: The RegCM accurately reflects the daily impact of meteorological factors on pollen emissions. Key physical processes, such as advection, convection, and wet and dry deposition, play essential roles in simulating the atmospheric dispersion and deposition of pollen.



This study validated the accuracy and reliability of the optimized emission potential
models for three pollen types using RegCM, effectively describing the daily variations
in pollen concentrations influenced by meteorological factors and key physical
processes. Furthermore, the pollen emissions model developed in this study can be
applied to other regions, offering potential for wider application. These
comprehensive results provide essential scientific support for pollen monitoring,
allergy prevention, and the selection of urban greening plants. Future research can
extend these methods and findings to larger-scale pollen emissions simulations and
forecasts, enhancing responses to pollen-related public health issues.
(5) Limitations and Future Prospects: Despite significant progress in
constructing the pollen emissions model and analyzing the spatiotemporal distribution
of pollen concentrations, some limitations persist. For broader application, more
extensive observation stations are needed to verify the model's accuracy, considering
the limited spatiotemporal resolution of current pollen concentration data. Simulating
specific species' pollen concentrations requires detailed plant functional type
distributions, which significantly impact the spatial distribution of pollen emissions
potential. The current research utilizes static plant functional type data, but dynamic
data would better reflect the impact of land use changes on pollen climates over
various temporal and spatial scales. Additionally, the complex relationship between
meteorological factors and pollen concentrations suggests that future research could
introduce more environmental and meteorological variables and apply advanced
machine learning algorithms to enhance the model's predictive capability.
In conclusion, This study successfully constructed a pollen emissions potential
model, systematically analyzed the spatiotemporal distribution of different pollen
types in autumn in Beijing, and explored their relationship with meteorological factors.
The model's accuracy and stability were validated using the RegCM, yielding notable
research results. Future research can further validate and extend this approach on a
larger scale and with higher resolution, providing comprehensive scientific support
for ecological environment protection and public health.



## Data availability

Meteorological data were sourced from the China Surface Climate Daily Dataset (https://data.cma.cn/data/cdcindex/cid/f0fb4b55508804ca.html), which requires appropriate permissions for access. Pollen data were provided by the Beijing Meteorological Bureau, and the authors do not have permission to share this data.

## Authorship contributions

**JL** performed the analysis, investigation, methodology, software development, validation, and original draft preparation. **XA** conceptualized the paper, provided resources, acquired funding, and conducted the review and editing. **ZS** and **CY** contributed resources, visualization, and data curation. **HQ**, **YZ**, and **ZL** were involved in visualization. All authors contributed to manuscript revisions.

### Declaration of competing interest

The authors declare that they have no known competing financial interests or personal relationships that could have appeared to influence the work reported in this paper.

## Acknowledgments

This work was supported by the National Key Research and Development Program of China (grant number 2022YFC3701205), Science and Technology Development Fund of the Chinese Academy of Meteorological Sciences (grant number 2023Z026) and the National Natural Science Foundation of China (grant number 41975173).

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
