# Peer review of "Construction and Application of a Pollen Emissions Model based on Phenology and Random Forests"

_EGUsphere, 2024_

## Author Comment (AC1)

This work by Li et al., has successfully constructed a pollen emission model based on phenology and random forest algorithm based on the 16 years of pollen observation data observed in Beijing. Moreover, the impact of temperature and relative humidity on the pollen emission has been reported. Furtherly, this pollen emission model was integrated into the regional climate model to validate the reliability of this model. The work presents a useful pollen prediction model for future prediction.

Response : Thank you for your positive feedback and valuable comments on our research work. We are very pleased to hear that you appreciate the pollen emission model we constructed based on pollen observation data in Beijing, as well as our study on the impact of temperature and relative humidity on pollen emissions. The following is a detailed response to your feedback

General points:

1. why the data was stopped at 2021? Could this model be used to predict those for year 2022, 2023?

Response 1: Thank you very much for your insightful and valuable suggestions. Our study developed a pollen emission model based on 16 years of pollen observation data (2006 – 2021). The model construction utilized pollen phenological information, including sDOY, eDOY, and ef. Daily monitoring data were used to validate the simulated pollen emissions against actual observations. Therefore, we believe that the pollen data for this period are sufficient to confirm the reliability of our emission model.

Additionally, pollen data in China are extremely scarce, and at present, we only have access to data up to 2021. For this reason, we have not made predictions for future years such as 2022 and 2023. However, our model is designed to be generalizable and can indeed be used to predict pollen emissions for future years. The reason we have not conducted this work is the lack of pollen data from future years to compare against. Your suggestion is very meaningful, and moving forward, we will actively seek to obtain more recent pollen data and conduct future predictions using our

established model.

2. there are too many figures and tables in this manuscript. I would suggest the authors to move some of them into the supporting information.

Response 2: Thank you for your suggestion. We agree with your point, and in response, we have moved some of the figures and tables from the main manuscript to the Supplementary Information section. Specifically, we have retained only one representative figure or table of the same type in the main manuscript, while the others have been moved to the Supplement. In particular, we have relocated Table 3-4 from the main manuscript to the Supplement, now labeled as Table S1-2. Additionally, Fig. 6, 7, 9, 10, 14, 15, 17, and 18 have been moved to the Supplement, now labeled as Fig. S5-S12.

We are confident that these adjustments will not affect the readability of the paper, and this adjustment will streamline the manuscript and improve its clarity while maintaining the essential information in the Supplementary section.

3. previous literatures on pollen prediction model should be introduced in the introduction. How about the advancement of the current model when compared to those in the literatures?

Response 3: Thank you for your valuable feedback. We have addressed your suggestion by incorporating a comprehensive overview of previous literature on pollen prediction models in the introduction. In this section, we highlighted the limitations of earlier studies, emphasizing the need for further research in this area.

We introduced a pollen phenology model to mitigate uncertainties associated with simulating the start and end of the pollen season (sDOY and eDOY). Additionally, we incorporated a RF algorithm to achieve a more precise estimation of annual pollen production. These enhancements represent a significant advancement in the quantitative accuracy of pollen emission studies compared to previous literature.

Furthermore, we integrated our final pollen emission simulations into the RegCM model, thereby significantly improving the accuracy of emission estimates. This

approach not only addresses the gaps identified in prior research but also contributes to a more robust understanding of pollen dynamics and their implications.

Minor points:

Line 31: RegCM. The first time it appeared, the full name should be appeared first. Similar for sDOY, eDOY in lines 134-135.

Response: Thank you for your suggestion. We have added the full name for RegCM where it first appears in the manuscript, as well as for sDOY and eDOY in lines 134-135.

Lines 58-62: references should be provided.

Response: Thank you for your suggestion. We have added the relevant references to the corresponding section in lines 58-62.

Line 95, Line 103: what is the difference between numerical model and pollen emission model? It would be better to clearly explain it here.

Response: Thank you for your valuable feedback. We have clarified the difference between numerical models and pollen emission models in the revised manuscript.

A numerical model is a mathematical framework used to simulate various physical processes through numerical approximations. It typically encompasses a broad range of applications, including atmospheric dynamics, climate systems, and pollutant dispersal. In contrast, a pollen emission model specifically focuses on estimating the release and distribution of pollen into the atmosphere. This model takes into account specific factors such as pollen phenology, vegetation types, and environmental conditions to estimate pollen emissions.

We have updated the manuscript to include this distinction in lines 95 and 103, providing a clearer understanding of how these models interact and complement each other.

Line 123: how was this formula obtained? Was this a novel formula or from literatures?

Response: Thank you for your question. We apologize for any confusion. The formula in question was developed based on existing pollen emission simulation formulas from previous research. It is an extension and refinement of established methodologies rather than a novel formula. We have added relevant references to the revised manuscript to clarify the basis of this formula. Thank you again for your valuable suggestion.

Line 147: what is the meaning of $Rs_1$, $Rs_2$, $Rs_{sig}$?

Response: Thank you for your question. $Rs_1$, $Rs_2$, and $Rs_{sig}$ represent three distinct autumn phenology models for simulating the start day (sDOY) and end day (eDOY) of autumn pollen. These models are based on temperature and SSH, and they are derived from the work of Meier & Bigler (2023). We adopted their approach and simplified it into these three formula categories to optimize the phenological simulations for pollen in the Beijing area.

$Rs_1$ is a model where pollen accumulation starts when both temperature and SSH fall below their respective thresholds ($T_{base}$ and $L_{base}$), and accumulation continues until a threshold is exceeded; $Rs_2$ follows a similar structure to $Rs_1$, but incorporates an inverse SSH factor in the accumulation process; $Rs_{sig}$ is based on a sigmoid function, where accumulation follows an exponential pattern based on temperature and SSH.

These three models serve as simplified yet comprehensive representations of autumn pollen phenology and are specifically used to optimize pollen simulations under local conditions. In this study, we utilized a simulated annealing algorithm to adjust key parameters (Y, $T_{base}$, $L_{base}$, x, y, start_day) and achieve the best fit for the model.

Line 182: Has the RF algorithm been used for pollen emission simulation in the previous literatures?

Response: Thank you for your insightful question. Indeed, the RF (Random Forest) algorithm has been utilized in previous studies to simulate daily pollen emissions. For instance, Valipour Shokouhi et al. (2024) successfully applied the RF algorithm by integrating meteorological variables and satellite data (MODIS, Landsat) to simulate

daily pollen concentrations in Switzerland, achieving favorable results. The RF algorithm is a widely applied machine learning technique with strong capabilities for handling nonlinear relationships and complex datasets. It is particularly effective in scenarios where traditional linear models may fall short due to the complexity of environmental variables involved.

In prior studies, the estimation of annual pollen production has often relied on empirical approaches or linear fitting based on the previous year's annual average temperature (PYAAT) (Wozniak and Steiner, 2017; Zhang and Steiner, 2022). However, our study found a very low correlation between PYAAT and $P_{annual}$ (as shown in Fig. 1), indicating that such approaches may not be suitable for simulating annual pollen production in the Beijing area.

Therefore, we explored machine learning algorithms and applied the RF algorithm to simulate annual pollen production based on meteorological factors. The simulation results significantly outperformed those of empirical algorithms. In our study, this is the first time that the RF algorithm has been used to simulate pollen production, and it has shown promising results. Similarly, the empirical simulations of start and end dates (sDOY, eDOY) based on PYAAT did not meet our expectations (Fig. 2), leading us to adopt the autumn phenology model for more accurate simulations.

[Figure]

Fig. 1 The relationship between autumn $P_{annual}$ and PYAAT in Beijing area

[Figure]

Fig. 2 The relationship between DOY and PYAAT in Beijing area

Section 3.1.1: was temperature the only variable in this autumn phenology model?

Response: Thank you for your valuable feedback. We would like to clarify that temperature was not the only variable considered in our autumn phenology model. In addition to temperature, we also took sunshine hours (SSH) into account. Specifically, in equations (4) to (6) of the manuscript, $T_i$ and $L_i$ represent the temperature and sunshine hours(SSH) on a given day, respectively, while $T_{base}$ and $L_{base}$ denote the thresholds for temperature and SSH. Detailed descriptions can be found in section 2.1.2 of our manuscript.

Moreover, we examined not only the effects of temperature and SSH, but also the impacts of different temperature types (TEM_Avg, TEM_Max, and TEM_Min) on the autumn phenology model. This approach allowed us to explore how varying temperature types influence different autumn phenological patterns, ultimately aiming to identify the optimal model for simulating pollen sDOY and eDOY in the Beijing region.

Lines 548-551: please provide more details or discussions regarding the linkage between pollen amount and climate change.

Response: Thank you for your insightful feedback regarding the connection between pollen production and climate change.

In our analysis, we observed that the interannual variations in pollen production are influenced by several climatic factors, including temperature, precipitation, and

sunshine hours. These elements can affect the growth cycles of pollen-producing plants, ultimately impacting their annual production levels. In response, we have added further discussion on the connection between climate change and pollen production in the original text.

Additionally, in the subsequent section, we delve into the relationships between specific meteorological factors and pollen production. This analysis provides a more detailed examination of how climate change could potentially alter pollen phenology and production patterns, further supporting the linkage between pollen amounts and climate trends discussed in our study.

Lines 617-625: it seems that basic information regarding the RegCM should be in the introduction section. It might be better for the authors to concise the current introduction and move such information into the introduction also.

Response: Thank you for your constructive feedback. We appreciate your suggestion to include basic information about the RegCM in the introduction section. We agree that this will help provide a clearer context for our study.

In our revised manuscript, we will work to concisely present the current introduction while integrating the relevant details about the RegCM. This will enhance the reader's understanding of the model and its significance to our research.

Section 3.3.1: some of them should be in the introduction section, and some may be better be in the method section.

Response: Thank you for your insightful feedback. We have carefully considered your suggestions and have made the following adjustments to the manuscript. The content from the first paragraph of section 3.3.1 has been moved to the introduction section to provide essential context for our study. Additionally, the remaining information regarding the RegCM model has been relocated to section 2.1.4, where we discuss the model in detail.

We believe these changes will enhance the clarity and organization of the manuscript. Thank you once again for your valuable input, and we look forward to any further

comments you may have.

---

## Author Response (AR2)

This work by Li et al., has successfully reported a pollen emissions model, which is of significance to study the impact of pollen on human health. The authors responded well to the comments of the previous two reviews. Moreover, the authors are very sincere and acknowledges the limitations of the paper, which they only have data up to 2021 and have not arbitrarily expanded the research conclusions. Looking forward to further data to develop the pollen model in the future. I suggest accepting this paper.

Response : Thank you very much for your positive and constructive comments on our manuscript. We are glad that you found our work on the pollen emissions model to be of significance, especially in understanding the impact of pollen on human health.

We sincerely appreciate your acknowledgment of how we responded to the comments from the previous reviews. We are committed to continuously improving the model and look forward to incorporating future data to enhance its accuracy and applicability.

We are grateful for your suggestion to accept the paper, and we will continue to work diligently on further developing the pollen model. Your thoughtful feedback is invaluable to us.

minor point:

1. Line 112, line 116, add full name about "KAMM/DRAIS" and "SILAM".

Response : Thank you for your suggestion. We have added the full name for "SILAM" where it first appears in the manuscript. However, regarding the KAMM/DRAIS model, we would like to mention that it is an abbreviation from German. The full name of the model, in its original language, is not provided in the references we cited. As such, we have opted not to include the full German name of the model in the manuscript.

2. Line 125, "CO2", subscript.

Response : Thank you for your suggestion. We have already made modifications.

3.  Add brief about the RegCM model into section 2.1.4.

Response : Thank you for your valuable suggestion. We fully understand the importance of providing an introduction to the RegCM model. However, we have included a brief description of the model in the second-to-last paragraph of the abstract (Lines 134 – 146). In this section, we highlight the significance of pollen models and explain the rationale for selecting the RegCM model. Given this consideration, we would prefer to keep the introduction to the RegCM model in the abstract rather than repeating it in Section 2.1.4. We believe this approach maintains the manuscript's structure and avoids redundancy. We appreciate your understanding and thoughtful feedback.

4.  Line 447-450, Table 2, Add unit for RMSE.

Response : Thank you for your insightful comment. We will update our manuscript accordingly to include this unit for clarity.